# SEAL: Steerable Reasoning Calibration of Large Language Models for Free

**Runjin Chen**[1]* **Zhenyu Zhang**[1]* **Junyuan Hong**[1] **Souvik Kundu**[2] **Zhangyang Wang**[1]
[1]The University of Texas at Austin, [2]Intel
{chenrunjin, zhenyu.zhang, jyhong, atlaswang}@utexas.edu, souvikk.kundu@intel.com

## Abstract

Large Language Models (LLMs), such as OpenAI's o1-series have demonstrated compelling capabilities for complex reasoning tasks via the extended chain-of-thought (CoT) reasoning mechanism. However, recent studies (Fu et al., 2024; Wang et al., 2025) reveal substantial redundancy in the CoT reasoning traces, which not only increases inference latency but also negatively impacts model performance by diverting attention to unnecessary reasoning paths. To address this issue, we investigate the internal reasoning structures of LLMs and categorize them into three primary thought types: execution, reflection, and transition thoughts. Moreover, our analysis reveals that excessive reflection and transition thoughts are strongly correlated with failure cases and these thought categories exhibit clear separation in the latent space. Based on these, we introduce SEAL (**S**teerable r**EA**soning ca**L**ibration), a training-free approach that seamlessly calibrates the CoT process, improving accuracy while demonstrating significant efficiency gains. SEAL consists of an offline stage for extracting the reasoning steering vector in the latent space, followed by an on-the-fly calibration of the reasoning trace through representation intervention using the steering vector. Notably, the steering vector exhibits strong transferability across various tasks. Extensive experiments across multiple models (DeepSeek-R1-Distill and QwQ-32B-Preview) and benchmarks (Math500, GSM8K, LiveCodeBench) validate the effectiveness of SEAL, up to a 11% improvement in accuracy while reducing reasoning tokens by 11.8% to 50.4%. Our code is publicly available at https://github.com/VITA-Group/SEAL.

## 1 Introduction

Recent advancements in Large Language Models (LLMs) have demonstrated significant success in extending their capabilities beyond simple language understanding tasks to more complex reasoning tasks, such as mathematical problem-solving (AlphaProof & Alpha-Geometry, 2024; Ahn et al., 2024; Luo et al., 2023; Yuan et al., 2023), planning (Wang et al., 2023; Valmeekam et al., 2023), and code debugging (Xia et al., 2024; Zhong et al., 2024). The primary factor contributing to this success is the ability of LLMs to execute an extended chain-of-thought (CoT)(Wei et al., 2022) reasoning process, which emulates human-like cognitive problem-solving by dynamically scaling test-time computation. Notable examples include OpenAI's o1-series models(OpenAI, 2024) and its open-source counterparts (Guo et al., 2025; Team, 2024; Muennighoff et al., 2025). These models are designed to explore diverse reasoning strategies, reflect on their decisions, iteratively refine solutions, and rigorously verify correctness—closely emulating human cognitive processes.

However, such an extended reasoning process introduces significant inference overhead due to the auto-regressive nature of LLMs. The generation process is typically memory-bound, with the KV cache size growing linearly with sequence length, leading to increasingly expensive memory transfer latency and further exacerbating inference inefficiencies. Moreover, lengthy reasoning is not always necessary. Studies have shown that LLMs often determine

---

*Equal contribution.

the correct final answer early in the reasoning process but continue generating excessive and redundant thought sequences (Fu et al., 2024). Such inefficient long responses can even degrade final performance, as models may become trapped in redundant verification loops (Chen et al., 2024) or suffer from underthinking due to unnecessary reasoning detours (Wang et al., 2025).

*Can we identify and calibrate the flawed reasoning pathways in current LLMs?* In this paper, we first decompose the entire reasoning process into a structured sequence of consecutive thoughts and categorize them into three types: (i) execution thoughts, (ii) reflection thoughts, and (iii) transition thoughts. Our analysis reveals that LLMs allocate significantly more thoughts to samples where they fail to produce the correct answer. We further analyze the conceptual representations of each thought type and observe that they are highly distinguishable in the latent space of deep layers. Building on this analysis, we propose SEAL (**S**teerable r**EA**soning ca**L**ibration), a training-free approach to calibrate reasoning paths, achieving a win-win of both efficiency and capability. With SEAL, we first perform an offline extraction of the reasoning steering vector in the latent space using a small subset of the training data (around one hundred samples). Then, during inference, we dynamically adjust the hidden states through arithmetic operations with the steering vector, enabling efficient reasoning calibration.

Our key contributions are as follows: (i) We systematically study the reasoning process of O1/R1-like LLMs and identify three distinct types of thoughts that compose the overall reasoning flow. Further, our analysis reveals that these thought types are highly distinguishable in the latent space; (ii) We propose a training-free strategy, SEAL, that effectively calibrates the reasoning process, reducing token consumption per question while simultaneously improving accuracy. (iii) Extensive experiments across diverse models (DeepSeek-R1-Distill and QwQ-32B-Preview) and various challenging benchmarks (such as Math500, GSM8K, LiveCodeBench) demonstrate that SEAL achieves consistent accuracy improvement by up to 11%, along with 11.8% to 50.4% token savings.

## 2 Recognizing Reasoning Patterns in LLMs

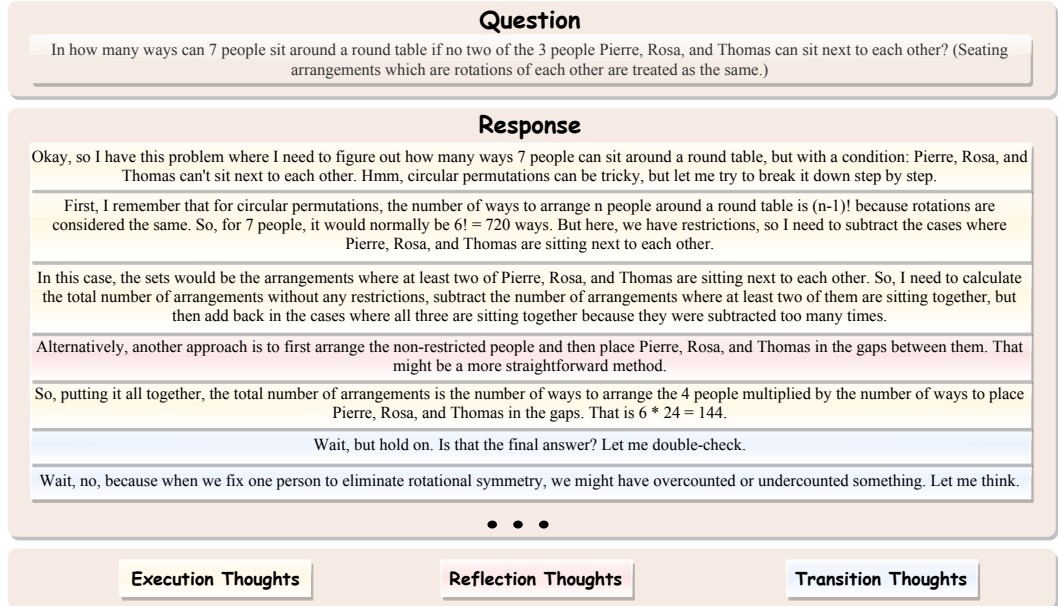

Figure 1: One example from DeepSeek-R1-Distill-Qwen-7B for a math question, where the entire response is divided into individual thought blocks.

We begin with a preliminary study on the reasoning patterns of O1-like LLMs that employ extended chain-of-thought (CoT) reasoning processes. To analyze the fine-grained reasoning steps, we observed that the model often uses "$\backslash n \backslash n$" to separate distinct steps in its responses. Therefore, we decompose the generated output O into a sequence of interconnected thoughts, segmenting each thought using two line break symbols, such that $O = (T_1, T_2, ..., T_N)$. Our experiments demonstrate that different thoughts can be categorized into distinct types, each exhibiting unique roles within the reasoning process. Figure 1 illustrates a representative example for a math question using DeepSeek-R1-Distill-Qwen-7B. We identify three types of thoughts: (i) Execution thoughts, where the model analyzes the problem and solves it step by step; (ii) Reflecting thoughts, where the model pauses the reasoning process to verify its steps; and (iii) Transition thoughts, where the model shifts its reasoning flow and rethink the problem from a different perspective.

Considering the statistical properties of various thoughts throughout the reasoning process, we present the results in Figure 2. Firstly, as the task difficulty increases, the model generates longer responses for correctly answering the questions, with the average token budget expanding from 1,534 to 3,323. This trend aligns with intuition, as more complex questions necessitate greater reasoning efforts for resolution, leading to longer response sequences. However, for samples of the same difficulty level, the number of thoughts in incorrect samples is significantly higher than in correct ones, with each type of thought exhibiting more steps in incorrect cases. Given that the difficulty of correct and incorrect samples is comparable, these results suggest that such excessive reasoning steps introduce significant redundancy beyond the necessary reasoning process and may negatively impact performance.

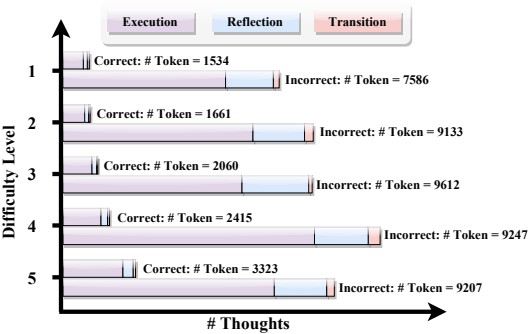

Figure 2: Statistics on the number of different types of thoughts for subsets of samples that the model answered correctly and incorrectly. The results are derived from DeepSeek-R1-Distill-Qwen-1.5B on the the Math-500 task. Response lengths are reported numerically.

Additionally, the increase in thoughts for incorrect samples is largely driven by a rise in reflection and transition thoughts, as each reflection or transition step is typically followed by several execution steps. As shown in Figure 9, this pattern aligns with recent studies, such as Chen et al. (2024), which demonstrate that allocating excessive reasoning steps to simple questions is ineffective. Similarly, Fu et al. (2024) finds that models often identify the correct answer early in the reasoning process but continue with redundant reasoning steps. Wang et al. (2025) argues that frequent transition thoughts disrupt the continuity of reasoning, leading to underthinking and performance degradation. Inspired by these studies, we summarize two major flaws in current O1-like reasoning process: (i) Efficiency: Frequent reflection and transition thoughts consume a significant token budget, introducing substantial efficiency overhead. Given that LLMs rely on auto-regressive generation and require increasing KV cache storage for long-context processing, excessive reasoning steps impose a considerable cost for computation and memory transfer. (ii) Effectiveness: Reflection and transition thoughts can lead to either overthinking or underthinking, diverting the model from the essential reasoning path. This distraction results in suboptimal performance, as the model fails to focus on the most direct and necessary reasoning steps.

In the following section, we aim to analyze the roles of different thoughts in the latent space and explore a controllable approach to mitigate redundant reflection and transition thoughts, thereby enhancing both the efficiency and effectiveness of the reasoning process.

## 3 Different Reasoning Patterns are Distinguishable in the Latent Space

To analyze the underlying mechanisms of different reasoning patterns, one challenge lies in the diversity of tokens present in various thoughts within the same category. Specifically, for all thoughts classified as reflection thoughts, the specific tokens in each instance exhibit substantial variation, as they are closely tied to the particular details of the question, as illustrated in Figure 1. This variability complicates detailed analysis. Meanwhile, transformer models process input sequences in a layer-wise manner, with representations at each layer capturing and accumulating conceptual knowledge rather than token-specific content. These latent space representations offer a meaningful way to understand reasoning dynamics. Thus, we conduct a preliminary study to analyze how different reasoning patterns are structured within the latent space.

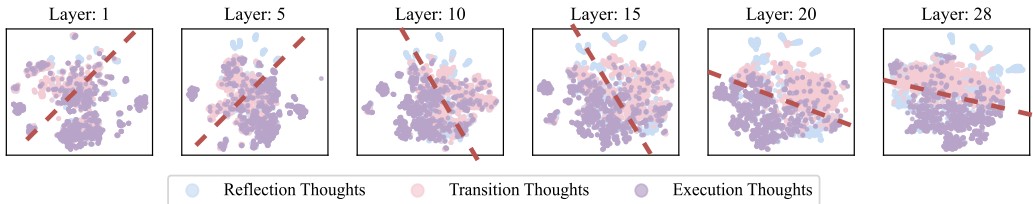

Figure 3: Results of t-SNE visualization of different reasoning thoughts in the latent space.

Experiments are conducted using DeepSeek-R1-Distill-Qwen-1.5B (Guo et al., 2025) on the Math-500 (Lightman et al., 2023) task. We first execute the general inference process and collect the representations $H_i$ corresponding to the "$\backslash n \backslash n$" of each thought in layer $i$. These representations encapsulate the entire information of each thought and determine the initial token of the subsequent thought, making them a strong indicator for analyzing the conceptual behavior of different thoughts. We then apply T-distributed Stochastic Neighbor Embedding (t-SNE) (Van der Maaten & Hinton, 2008) to project $H_i$ into a two-dimensional space. The results are presented in Figure 3, reveal several key observations: (i) Execution thoughts are clearly separable from non-execution thoughts (i.e., reflection and transition thoughts) in the latent space. For example, at Layer 20, execution thoughts exhibit almost no overlap with other types of thoughts. (ii) The separability of different thought is significantly better in deep layers, whereas the initial layers struggle to distinguish them. This aligns with the expectation that shallow layers primarily capture low-level features, while deeper layers encode more abstract conceptual and semantic knowledge (Liu et al., 2024; Jin et al., 2024). (iii) Reflection and transition thoughts are more similar to each other than to execution thoughts. Intuitively, both reflection and transition thoughts involve reconsidering or modifying previous reasoning steps, whereas execution thoughts represent the original step-by-step reasoning process. Based on these experiments, different reasoning patterns are qualitatively distinguishable in the latent space.

## 4 Steerable Reasoning Calibration

Based on our preliminary investigation, we propose SEAL, a training-free framework designed to achieve **S**teerable r**EA**soning ca**L**ibration. As illustrated in Figure 4, SEAL comprises two stages: the off-the-shelf extraction of the reasoning steering vector and the on-the-fly intervention in the latent space during decoding. The core insight behind SEAL is to identify the steering vector that controls the ratio of reflection and transition thoughts. And use that vector to effectively reduce redundant token usage, achieving a win-win of efficiency and accuracy.

### 4.1 Extraction of Reasoning Steering Vector

**Collecting Reasoning Processing.** We begin by using a validation set comprising samples from reasoning benchmarks. In our experiments, we by default to use a selected set of

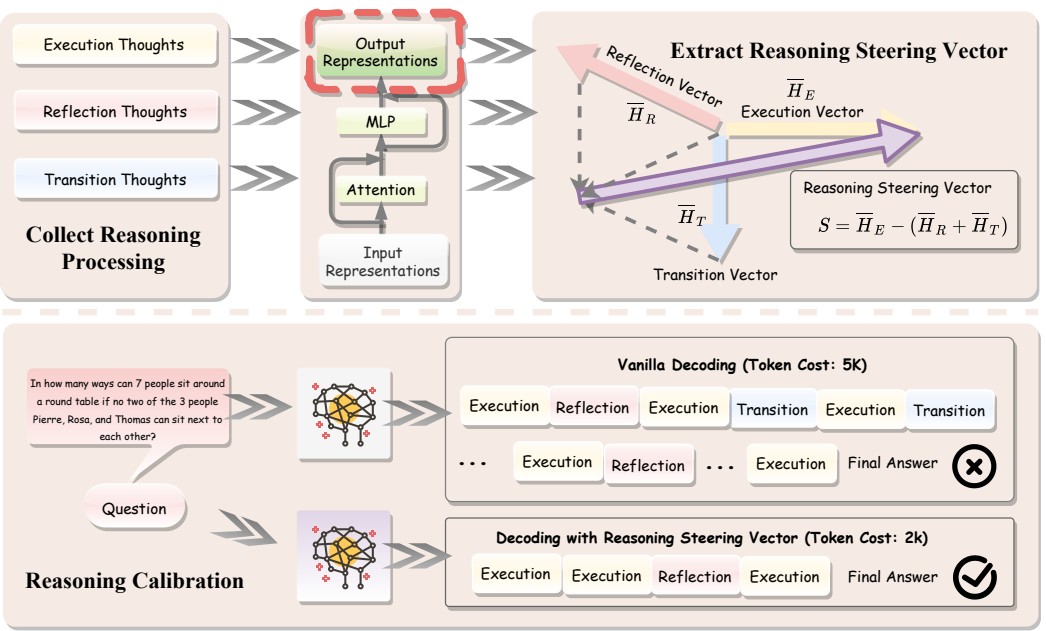

Figure 4: Overview of our SEAL framework. The upper subfigure illustrates the offline extraction process of the reasoning steering vector, while the lower subfigure depicts the inference process utilizing the extracted steering vector.

1000 training samples from the training set of the Math (Hendrycks et al., 2021) dataset and performed inference using the target model, such as DeepSeek-R1-Distill-Qwen-7B, to generate the whole reasoning process for each sample. Similar to Section 3, we segmented the reasoning process into a set of individual thoughts using a line-breaker symbol. Each thought was then categorized into one of three predefined classes: execution, reflection, or transition thoughts. The classification was performed through keyword identification. For example, if a thought contained keywords like 'Alternatively,' it was categorized as a transition thought. The complete list of keywords used to classify transition and reflection thoughts is provided in Appendix B. Any remaining thoughts were classified as execution thoughts.

**Calculating Steering Vector.** For each thought $j$, we extract the output representations from the $i^{th}$ transformer blocks corresponding to the first token "$\backslash n \backslash n$", denoted as $H_i^j$. We then compute the average representations for each thought category:

$$\overline{H}_{C,i} = \frac{1}{N_C} \sum_{j \in C} H_i^j, \quad C \in \{\text{Execution}, \text{Reflection} \cup \text{Transition}\}$$

Based on the results in Section 2, the primary objective of SEAL is to mitigate unnecessary reflection and transition thoughts while preserving essential execution thoughts. To achieve this, we compute the reasoning steering vector $S$ as the arithmetic combination of different thought vectors, *i.e.*,

$$S = \overline{H}_E - \overline{H}_{RT}$$

## 4.2 Decoding with Latent Space Intervention

We then perform on-the-fly calibration of the reasoning process using the reasoning steering vector $S$. Specifically, during the decoding process, at the end of each thought, we intervene in the representations of all $\backslash n \backslash n$ tokens by applying an offset derived from $S$, formulated as: $\widetilde{H} = H + \alpha \cdot S$, where $\alpha$ is a hyperparameter controlling the strength of the intervention. Since the extraction process is conducted offline, it does not introduce any additional latency

overhead during decoding. Furthermore, the computational cost of latent space intervention is negligible, compared with the original forward pass.

For hyperparameter selection, we conduct a series of ablation studies in Section 5.3.1 and, by default, set $\alpha = 1.0$ and the intervention layer as 20 for Deepseek-R1-Distill-Qwen-1.5B and Deepseek-R1-Distill-Qwen-7B, 55 for QwQ-32B-Preview. Additionally, we find that the validation set used for extracting the steering vector generalizes well across different tasks, such as transferring from mathematical reasoning to code generation. Further results are provided in Section 5.2.1

# 5 Experiments

## 5.1 General Setup

We evaluate the effectiveness of our SEAL with several popular reasoning models, including Deepseek-R1-distill-Qwen-1.5B, Deepseek-R1-distill-Qwen-7B (Guo et al., 2025) and QwQ-32B-Preview (Team, 2024; Yang et al., 2024a). And we conduct the experiments on four datsets: (i) **Math500** (Hendrycks et al., 2021): A challenging math dataset comprising problems from high school math competitions. We adopt a subset of 500 problems selected by OpenAI[1] as test set. Additionally, the Math dataset categorizes problems by difficulty on a scale from 1 to 5. We designate the subset of 500 problems with difficulty levels 4 or 5 as the hard test set. (ii) **GSM8k** Cobbe et al. (2021): A dataset of high-quality problems at the grade school math level. This dataset exhibits high linguistic diversity while relying on relatively simple grade school math concepts. The test set consists of 1,319 problems. (iii) **LiveCodeBench** Jain et al. (2024): A dataset containing 400 Python coding problems released between May 2023 and March 2024, each accompanied by a set of test samples for verifying program correctness.

## 5.2 End-to-End Results

Table 1: Comparison results on the Math500 task. The reasoning steering vector of SEAL is derived from a subset of training samples from the Math500 task.

| Methods | Math-500 | | Math-500 (Hard) | |
|---|---|---|---|---|
| | Accuracy@1 (↑) | #Tokens (↓) | Accuracy@1 (↑) | #Tokens (↓) |
| Deepseek-R1-Distill-Qwen-1.5B | 67.0 | 4526 | 54.2 | 5737 |
| *w.* Logits Penalty$_{\text{Transition}}$ | 68.2 (+1.2) | 3660 | 55.7 (+1.5) | 4766 |
| *w.* Logits Penalty$_{\text{Reflection}}$ | 75.0 (+8.0) | 4416 | 65.3 (+11.1) | 5644 |
| *w.* Logits Penalty$_{\text{Both}}$ | 76.6 (+9.6) | 3340 | 63.7 (+9.5) | 4552 |
| *w.* SEAL | **78.0 (+11.0)** | **3154** | **68.3 (+14.1)** | **4086** |
| Deepseek-R1-Distill-Qwen-7B | 85.8 | 3389 | 79.8 | 4176 |
| *w.* Logits Penalty$_{\text{Transition}}$ | 86.2 (+0.4) | 3264 | 79.4 (-0.4) | 4108 |
| *w.* Logits Penalty$_{\text{Reflection}}$ | 88.2 (+2.4) | 2905 | 81.7 (+1.9) | 3773 |
| *w.* Logits Penalty$_{\text{Both}}$ | 87.4 (+1.6) | 2807 | 80.9 (+1.1) | 3596 |
| *w.* SEAL | **89.4 (+3.6)** | **2661** | **84.0 (+4.2)** | **3365** |
| QwQ-32B-Preview | 90.4 | 2113 | 86.6 | 2793 |
| *w.* Logits Penalty$_{\text{Transition}}$ | 90.2 (-0.2) | 2115 | 85.5 (-1.1) | 2820 |
| *w.* Logits Penalty$_{\text{Reflection}}$ | 88.8 (-1.6) | 2104 | 84.0 (-2.6) | 2778 |
| *w.* Logits Penalty$_{\text{Both}}$ | 89.4 (-1.0) | 1966 | 83.2 (-3.4) | 2681 |
| *w.* SEAL | **91.0 (+0.6)** | **1716** | **85.5 (-1.1)** | **2323** |

---

[1]https://huggingface.co/datasets/HuggingFaceH4/MATH-500

Table 2: Generalization results of SEAL across different datasets. The reasoning steering vector is obtained from a subset of training samples from the Math500 task.

| Methods | GSM8K | | LivecodeBench | |
|---|---|---|---|---|
| | Accuracy@1 (↑) | #Tokens (↓) | Accuracy@1 (↑) | #Tokens (↓) |
| Deepseek-R1-Distill-Qwen-1.5B | 74.1 | 2015 | 18.5 | 8205 |
| w. Logits Penalty$_{Transition}$ | 75.2 (+1.1) | 1862 | 16.8 (-1.7) | 8180 |
| w. Logits Penalty$_{Reflection}$ | 78.5 (+4.4) | 1214 | 26.0 (+7.5) | 7160 |
| w. Logits Penalty$_{Both}$ | 78.3 (+4.2) | 1100 | 25.8 (+7.3) | 7050 |
| w. SEAL | **82.0 (+7.9)** | **999** | **28.5 (+10.0)** | **6923** |
| Deepseek-R1-Distill-Qwen-7B | 88.0 | 1142 | 44.5 | 6856 |
| w. Logits Penalty$_{Transition}$ | 88.6 (+0.6) | 1058 | 45.8 (+1.3) | 6697 |
| w. Logits Penalty$_{Reflection}$ | 87.9 (-0.1) | 904 | 44.0 (-0.5) | 6166 |
| w. Logits Penalty$_{Both}$ | **88.6 (+0.6)** | 885 | 46.3 (+1.8) | 6065 |
| w. SEAL | 88.4 (+0.4) | **811** | **51.7 (+7.2)** | **5974** |
| QwQ-32B-Preview | 95.4 | 698 | 62.5 | 6016 |
| w. Logits Penalty$_{Transition}$ | 95.1 (-0.3) | 657 | 63.5 (+1.0) | 5856 |
| w. Logits Penalty$_{Reflection}$ | **95.8 (+0.4)** | 664 | 63.0 (+0.5) | 5847 |
| w. Logits Penalty$_{Both}$ | 95.0 (-0.4) | 644 | 63.0 (+0.5) | 5847 |
| w. SEAL | 95.7 (+0.3) | **525** | **63.5 (+1.0)** | **5309** |

### 5.2.1 Baseline and Evaluation Metrics.

We compare SEAL with another training-free method, namely logits penalty Wang et al. (2025). Following their setup, we reduce the logits of tokens associated with characteristic reflection or transition words, such as 'wait' and 'alternatively.' The adjustment values were selected from {-1, -3, -10, -20}, with -3 generally yielding the best results. For each task, we report both the average accuracy and token count across the test set to assess the effectiveness and efficiency of SEAL. To ensure reproducibility, we use greedy search during decoding. Results using sampling are additionally reported in the appendix E.

### 5.2.2 Main Results.

**SEAL demonstrates superior performance and significant efficiency gains.** As shown in Table 1 and 2, across diverse models and tasks, SEAL not only improves accuracy but also reduces response length, enhancing reasoning efficiency. For example, SEAL enhances performance by up to 14.1% in accuracy while reducing token usage by 28.8% on hard problems in the Math-500 benchmark. This improvement stems from the model's ability to avoid excessive rechecking and minimize unnecessary transition thoughts after steering, leading to a more direct path to the final answer. More results on sequence length comparisons can be found in Appendix F. Additionally, SEAL prevents the model from becoming trapped in an endless cycle of rechecking and switching, further optimizing the reasoning process.

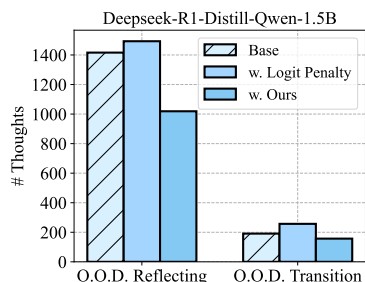

Figure 5: Logits Penalty makes other reflection and transition thoughts increase

**The reasoning steering vector exhibits high transferability across different tasks.** We applied the steering vector extracted from the MATH dataset to GSM8K (out-of-distribution samples within the same domain) as well as LiveCodeBench (transferring to a different domain). SEAL consistently enhanced both performance and efficiency across all datasets (As shown in Table 2), with accuracy improvements ranging from 0.3% to 10.0%, while reducing token usage by up to 50.4%. This indicates that the underlying patterns of rechecking and switching thoughts are

generalizable across different tasks and domains. Consequently, our method can be effectively applied to new domains without incurring the additional computational overhead of extracting a new steering vector for each task.

**Latent space calibration is more effective than token-space adjustment**. One limitation of the Logit Penalty (*i.e.*, token-space adjustment) method is that it typically operates on individual tokens, such as alternatively or wait, rather than adjusting at the conceptual level. However, reflection and transition thoughts are often conveyed through phrases or longer sentences, such as let me double-check or another approach is. These cases are difficult to suppress using Logit Penalty alone. Furthermore, we observed that even after lowering the logits of representative tokens for these thoughts, the model still exhibited a tendency toward reflection and transition—albeit in a more implicit manner, through rephrased expressions. To quantify this effect, we measured the number of reworded reflection and transition steps. As shown in Figure 5, logit penalty led to an increase in such steps. In contrast, our steering method suppresses the entire reflection/transition concept rather than just specific tokens. As a result, these steps were significantly reduced, demonstrating that steering is a stronger and more effective approach than logit penalty.

### 5.2.3 Quantitative Evaluation of Efficiency

Table 3: End-to-end efficiency comparison on the Math500 task benchmark. RD denotes the relative reduction ratio compared to the baseline method. Both the average and maximum reduction ratios are reported.

| Models | Throughput (#Token/second) | #Tokens/Sample Avg. RD (%) | Max. RD (%) | Avg. | Time/Sample (second) Avg. RD (%) | Max. RD (%) | Avg. |
|---|---|---|---|---|---|---|---|
| Deepseek-R1-Distill-Qwen-1.5B | 41.4 | N/A | N/A | 4666.9 | N/A | N/A | 112.7 |
| *w.* SEAL | 43.5 | 34.70 | 80.46 | 3047.7 | 37.89 | 83.65 | 70.0 |
| Deepseek-R1-Distill-Qwen-7B | 37.2 | N/A | N/A | 3522.1 | N/A | N/A | 94.6 |
| *w.* SEAL | 39.5 | 28.78 | 81.26 | 2508.6 | 32.88 | 86.61 | 63.5 |

We report the end-to-end efficiency results in Table 3 using 50 random samples in Math500 benchmark. Without loss of generality, our implementation is based on the Hugging Face Library with BF16 precision. Throughput and time cost are measured on a single NVIDIA GH200 GPU without offloading. Our observations indicate that the additional computation introduced by adding the steering vector is negligible. SEAL even achieves a slight improvement in throughput (*i.e.*, approximately 2 tokens per second) by eliminating the overhead of KV cache in extremely long sequences. Additionally, SEAL significantly reduces token consumption in responses, resulting in an average reduction in response time per query by 32.9% to 37.9%. Notably, the most improved sample achieves a reduction of 83.65% to 86.61%. These results further substantiate the efficiency gains and practical advantages attained through our approach.

## 5.3 Ablation Study

In this section, we present ablation studies on the effects of applying the steering operation across different reasoning types, layers, and strengths.

### 5.3.1 Ablation Study about the Steering Type

Table 4 presents the results of steering different reasoning types. We observe that weakening execution thoughts leads to performance degradation, as execution plays a crucial role in the reasoning process. In contrast, reducing the

Table 4: Ablation study of steering type. Experiments are conducted with Deepseek-R1-Distill-Qwen-7B

| Method | Math500 | GSM8K | LiveCodeBench |
|---|---|---|---|
| Baseline | 85.8 | 88.0 | 44.5 |
| SEAL (weakening Reflection) | 88.6 | 88.4 | 50.7 |
| SEAL (weakening Transition) | 87.8 | 88.5 | 49.0 |
| SEAL (weakening Execution) | 65.0 | 79.6 | 27.5 |
| SEAL | 89.4 | 88.9 | 51.7 |

influence of reflection and transition thoughts individually can still enhance performance. Moreover, enabling the steering vector to weaken both reflection and transition yields the most significant improvements. Consequently, the optimal reasoning formulation is given by $S = \overline{H}_E - \overline{H}_{RT}$.

### 5.3.2 Ablation Study about the Steering Layer

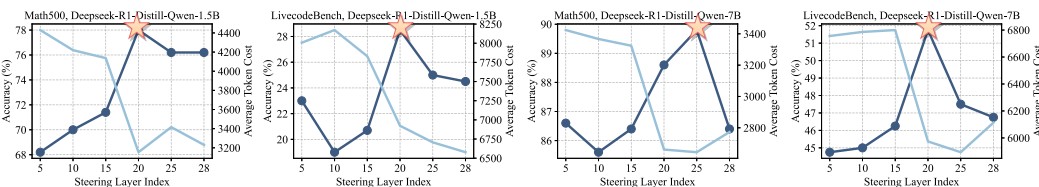

Figure 6: Ablation study of the steering layers. Zoom in for better visualization.

In this section, we aim to identify the optimal layer for steering. We extract reasoning steering vectors from different layers and apply corresponding steering to evaluate their effectiveness. The results, shown in Figure 6, indicate that mid-to-late layers yields the best performance, and the optimal layer shows strong generalization across different model and tasks. This aligns with the typical forward pass dynamics of LLMs: early layers focus primarily on token-level representations, while middle layers progressively merge information across tokens to form higher-level concepts Liu et al. (2024); Jin et al. (2024). In contrast, the final layers primarily summarize context to predict the next token. Consequently, concept-level steering tends to be more effective in the middle layers.

### 5.3.3 Ablation Study about the Steering Strength

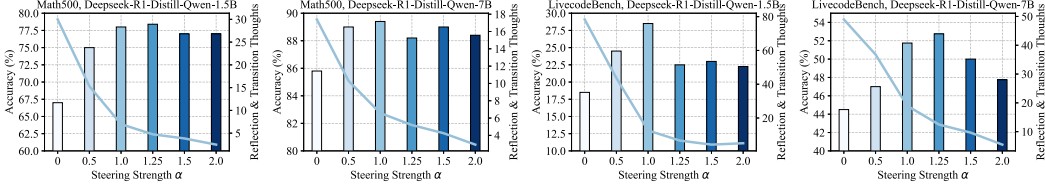

Figure 7: Ablation study of the steering strength. Zoom in for better visualization.

We further investigate the effect of steering strength on the final outcomes, with results presented in Figure 7. Our findings show that applying the reasoning steering vector effectively reduces reflection and transition thoughts. However, minimizing these thoughts does not always yield better performance. This suggests that a balanced number of reflection and transition steps can be beneficial for arriving at the correct final answer. In our experiments, a steering coefficient of $\alpha = 1.0$ consistently demonstrated strong performance across various tasks, and we maintained this setting throughout all our experiments.

## 6 Conclusion

In this paper, we investigate the internal reasoning dynamics of large language models (LLMs), revealing that excessive engagement in reflection and transition thoughts often introduces inefficiencies and errors during extended chain-of-thought reasoning. By categorizing reasoning steps into execution, reflection, and transition components, we show that these types exhibit clear and consistent separability within the latent space. This insight enables more precise control over the reasoning process. Leveraging this observation, we propose SEAL, a lightweight, training-free method that calibrates representations to steer reasoning trajectories. Specifically, SEAL adjusts hidden states on the fly using a precomputed steering vector, effectively promoting execution while mitigating the influence of

less productive thoughts. Our approach improves both reasoning efficiency and accuracy without requiring model fine-tuning or architectural modifications. Experimental results across various LLMs and benchmarks confirm the robustness and generality of SEAL. It not only reduces inference time but also consistently enhances performance, offering a practical and interpretable solution for optimizing reasoning in language models.

## Acknowledgment

Z. Wang is in part supported by NSF Awards 2523383 (AIMing), the NSF AI Institute for Foundations of Machine Learning (IFML), and an Intel single PI SRS gift funding.

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

# A Related Works

**Reasoning in Latent Space**. Some studies have found that LLMs inherently perform latent reasoning within their hidden computations Yang et al. (2024b); Shalev et al. (2024); Hao et al. (2024); Tack et al. (2025). Yang et al. (2024b) explores reasoning paths in multi-hop reasoning tasks and finds that intermediate entities can be recovered from latent representations. Shalev et al. (2024) shows that multiple parallel latent reasoning paths may exist in the middle layers. Collectively, these works demonstrate that LLMs naturally engage in reasoning within their internal latent representations, embedding reasoning concepts and cues. Additionally, several approaches have sought to further enhance and refine reasoning within the latent space of existing models Tack et al. (2025); Hao et al. (2024). These efforts collectively inspire further advancements in optimizing reasoning paths at the level of latent representations.

**Under-thinking and Overthinking**. Several studies have shown that current O1-like models exhibit both overthinking and underthinking issues in their reasoning paths Chen et al. (2024; 2025); Wang et al. (2025); Fu et al. (2024). Overthinking, as discussed in Chen et al. (2024), refers to allocating excessive computational resources to simple questions with minimal performance gains. Fu et al. (2024) further reveals that LLMs often determine the correct final answer early in the reasoning process but continue generating excessive and redundant thought sequences. Conversely, underthinking, as described in Wang et al. (2025), occurs when models frequently switch between thoughts without sufficiently exploring promising reasoning paths to reach a correct solution. Our work aims to identify these flawed reasoning patterns within the latent space and apply steering techniques to mitigate them.

**Representation Engineering (RepE)**. RepE Zou et al.; Wehner et al. (2025); Wu et al. (2025) places the model's latent representations at the core of analysis, identifying target concepts within these representations and steering them to control model behavior. These methods offer advantages such as efficiency, flexibility, and strong interpretability. Inspired by RepE, our work adopts a similar approach to mitigate flawed reasoning patterns in the model's reasoning path.

# B Criteria for Recognizing Reflection, Transition, and Execution Thought

## B.1 Criteria for different thoughts

Table 5: Criteria for recognizing reflection, transition, and execution thoughts

| | |
|---|---|
| **Transition** | **Prefix**: Alternatively |
| | **Phrase**: think differently, another way, another approach, another method, another solution, another strategy, another technique |
| **Reflection** | **Prefix**: Wait |
| | **Phrase**: verify, make sure, hold on, think again, 's correct, 's incorrect, Let me check, seems right |

Through observation, we have summarized a set of rules to determine whether a given step belongs to reflection, transition, or execution. These rules can be categorized into two main types.

The first type is the prefix rule, which relies on the observation that many transition or reflection thoughts often begin with words such as "wait" or "alternatively". If a step starts with these words, we classify it as a transition or reflection. However, this criterion alone is insufficient, as many steps do not meet this condition.

To address this limitation, we designed the phrase rule, which identifies common phrases that frequently appear in the middle of a step. As shown in Table 5, if any of these predefined phrases occur within a step, we classify it as a transition or reflection. Steps that do not meet either of these criteria are categorized as execution. During the classification process, letter case is ignored.

## B.2  Ablation on different criteria

Table 6: Ablation study of thoughts recognizing criteria with Deepseek-R1-Distill-Qwen-1.5B

| Methods | MATH-500 | | GSM | | LiveCodeBench | |
| --- | --- | --- | --- | --- | --- | --- |
| | Accuracy@1(↑) | Tokens(↓) | Accuracy@1(↑) | Tokens(↓) | Pass@1(↑) | Tokens(↓) |
| Base | 67.0 | 4526 | 74.1 | 2015 | 18.5 | 8205 |
| SEAL (w. Prefix Only) | 78.0 | 3115 | 81.7 | 999 | 25.3 | 7119 |
| SEAL (w. Phrase Only) | 75.6 | 3522 | 80.4 | 1210 | 23.5 | 7466 |
| SEAL | 78.0 | 3153 | 82.0 | 999 | 28.5 | 6923 |

We also conducted an ablation study on the criteria rules, evaluating the performance of using only the prefix rule or only the phrase rule to extract reflection and transition thoughts for steering. The results, presented in Table 6, indicate that even with only a subset of the rules, we can still achieve competitive results. This suggests that we do not need overly rigid pattern designs; as long as these rules capture key concept-level information about reflection and transition, we can obtain an effective steering vector. Naturally, more precise rules can further enhance performance.

## B.3  LLM-Based Labeling and Comparison with Heuristic Method

In addition to the keyword-based categorization introduced above, we also experimented with using a frontier model (GPT-4o) to label thought types. Given the full reasoning context and the current step, we prompted the LLM to classify each thought as *execution*, *reflection*, or *transition*.

Interestingly, our experiments showed that the keyword-based method achieves slightly better end-task performance in SEAL as shown in Table 7.

Table 7: Comparison of thought labeling methods on MATH500 and Deepseek-R1-7B-Distill.

| Method | Accuracy (%) |
| --- | --- |
| Baseline | 85.8 |
| SEAL (Keyword-based) | **89.4** |
| SEAL (LLM-based) | 88.6 |

That said, the LLM-based method remains appealing due to its **ease of adaptation to new domains**, **languages**, or **expanded categories**, and could be especially useful in scenarios where heuristic rules are hard to define or scale.

## C  SEAL Performance on In-Distribution Steering Vector

We also experimented with generating the steering vector using in-distribution data from GSM8K and LiveCodeBench. For GSM8K, we followed the same approach as in MATH-500, selecting 1000 samples from the training set to construct the steering vector. For LiveCodeBench, since we used the authors' released version1 as our test set, we leveraged 480 samples from the difference set between the later released version and the first version to collect thoughts and generate the steering vector.

As shown in Figure 8, steering vectors derived from in-distribution data proved effective but did not perform as well as those from MATH-500. This discrepancy can be attributed to the

Table 8: Performance of SEAL with a Reasoning Steering Vector Derived from In-Distribution Samples

| Methods | GSM8K | | LivecodeBench | |
|---|---|---|---|---|
| | Accuracy@1 (↑) | #Tokens (↓) | Pass@1 (↑) | #Tokens (↓) |
| *Deepseek-R1-Distill-Qwen-1.5B* | | | | |
| Base | 74.1 | 2015 | 18.5 | 8205 |
| SEAL | 80.9 | 1196 | 23.8 | 7218 |
| *Deepseek-R1-Distill-Qwen-7B* | | | | |
| Base | 88.0 | 1142 | 44.5 | 6856 |
| SEAL | 88.6 | 786 | 49.0 | 6015 |

relatively homogeneous nature of GSM8K and LiveCodeBench samples, whereas MATH-500 encompasses a wider range of difficulty levels and problem types, making it easier to obtain a more generalized steering vector. These results further validate our hypothesis that it is unnecessary to extract a new vector for each task individually, a well-curated, high-quality dataset is sufficient to generate a generalizable steering vector.

## D    Generalization to Open-Ended Planning Tasks

To further evaluate the generalizability of SEAL beyond structured domains such as mathematics and programming, we conducted additional experiments on a more open-ended reasoning benchmark: the **NaturalPlan** dataset( Zheng et al. (2024)), which focuses on *natural language calendar planning*.

Importantly, we reused the **reasoning steering vector extracted from the MATH domain**, to assess cross-domain transferability. As shown in Table 9, SEAL achieved consistent improvements in both accuracy and response length compared to the baseline, indicating that the latent patterns of reflection and transition overuse also appear in more open-ended planning contexts.

Table 9: Results on the calendar planning task from the NaturalPlan benchmark using Deepseek-R1-Distill-7B.

| Method | Accuracy (%) | Avg. Response Length |
|---|---|---|
| Baseline | 14.9 | 6341 |
| SEAL (Math Vector) | **18.6** | 3531 |

# E   Results with Sampling-Based Decoding

Table 10: Comparison results on the Math500 and LiveCodeBench. Using $t = 0.6$, $top\_p = 0.95$ for decoding

| Methods | Math-500 | | LiveCode | |
| | Accuracy@1 ($\uparrow$) | #Tokens ($\downarrow$) | Pass@1 ($\uparrow$) | #Tokens ($\downarrow$) |
|---|---|---|---|---|
| Deepseek-R1-Distill-Qwen-1.5B | 79.9 | 4021 | 29.6 | 7639 |
| *w.* Logits Penalty$_{Transition}$ | 80.9 (+1.0) | 3802 | 29.3 (-0.3) | 7567 |
| *w.* Logits Penalty$_{Reflection}$ | 78.8 (-1.1) | 3416 | 28.9 (-0.7) | 6727 |
| *w.* Logits Penalty$_{Both}$ | 80.3 (+0.4) | 3218 | 28.2 (-1.4) | 6755 |
| *w.* SEAL | **81.6 (+1.7)** | **2976** | **32.2 (+2.6)** | **6468** |
| Deepseek-R1-Distill-Qwen-7B | 89.7 | 3334 | 49.7 | 6609 |
| *w.* Logits Penalty$_{Transition}$ | 88.5 (-0.2) | 3318 | 48.7 (-1.0) | 6652 |
| *w.* Logits Penalty$_{Reflection}$ | 89.7 (+0.0) | 2942 | 53.1 (+3.4) | 5920 |
| *w.* Logits Penalty$_{Both}$ | 90.3 (+0.6) | 2831 | 53.8 (+4.1) | 5895 |
| *w.* SEAL | **91.3 (+1.6)** | **2695** | **56.3 (+6.6)** | **5871** |
| QwQ-32B | 91.1 | 3650 | 76.2 | 5610 |
| *w.* Logits Penalty$_{Transition}$ | 90.7 (-0.4) | 3547 | 75.3 (-0.9) | 5648 |
| *w.* Logits Penalty$_{Reflection}$ | 91.9 (+0.8) | 3270 | 81.2 (+5.0) | 5216 |
| *w.* Logits Penalty$_{Both}$ | 92.1 (+1.0) | **3150** | 81.5 (+5.3) | 5169 |
| *w.* SEAL | **92.5 (+1.4)** | 3160 | **82.8 (+6.6)** | **4959** |

We also experimented with decoding via sampling, using temperature $t = 0.6$ and $top\_p = 0.95$. The results are reported in Table 10, averaged over three runs. Similar to greedy decoding, we observe that SEAL achieves both improved performance and reduced token usage.

# F   Comparison of Generated Sequence Lengths

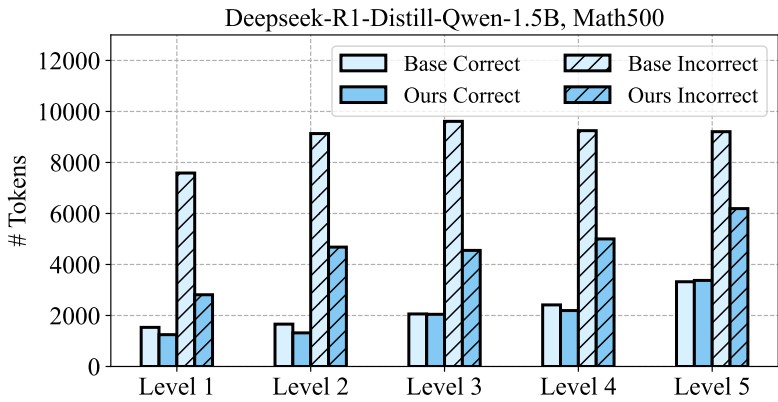

Figure 8: Comparison results of generated sequence lengths. We report the average sequence lengths for samples across different difficulty levels and separately for subsets where the model produces correct or incorrect answers. The experiments are conducted on the Math500 task using the DeepSeek-R1-Distill-Qwen-1.5B model.

As shown in Figure 8, applying SEAL significantly reduces sequence length for subsets where the model produces incorrect answers. This reduction occurs because the original reasoning process is often disrupted by unnecessary reflection and transition thoughts.

Mitigating these distractions allows the model to focus on the core reasoning path, enhancing performance while minimizing extra token consumption. Additionally, SEAL can slightly reduce redundant tokens even for correctly answered samples, further demonstrating its efficiency gains.

## G   Disentangling SEAL's Gains from Truncation Effects

While SEAL demonstrates performance improvements, one may question whether the gains stem primarily from *suppressing redundant thoughts* or simply from *shortening the sequence length*, which might reduce token-level errors. To disentangle these effects, we conducted a controlled comparison against a baseline that forcibly limits reasoning length without semantic steering.

Specifically, we introduced a fixed **"think budget"** mechanism, where the model is instructed to insert an </think> token once the total token count in the reasoning process exceeds a preset budget. After this point, the model proceeds directly to the answer summarization phase. We set the budget to 3500 tokens, closely matching the average output length of SEAL to ensure a fair comparison.

Table 11: Comparison between SEAL and simple truncation baseline on Deepseek-R1-Distill-7B (MATH500).

| Method | Accuracy (%) | Avg. Response Length |
|---|---|---|
| Baseline | 85.8 | 3389 |
| Baseline + Think Budget = 3500 | 85.0 | 2540 |
| SEAL | **89.4** | 2661 |

As shown in Table 11, the truncation-based baseline underperforms SEAL, despite producing similarly shorter outputs. This demonstrates that SEAL's advantage is not merely due to sequence shortening. Instead, it effectively **guides the model to suppress specific types of non-productive reasoning**, such as unnecessary reflection and transition thoughts, leading to better-calibrated and more accurate reasoning trajectories.

## H   Thoghts Patten Demonstration Examples

We present a reasoning path for a sample from MATH-500 in Figure 9. This sample is relatively easy, and the model is able to arrive at the correct answer within a few reasoning steps. However, instead of confidently settling on the correct solution, the model repeatedly attempts to verify and recheck its previously correct answer, continuously switching its thoughts. Ultimately, this leads the model into a loop, causing it to deviate from the correct reasoning path and produce an incorrect final answer.

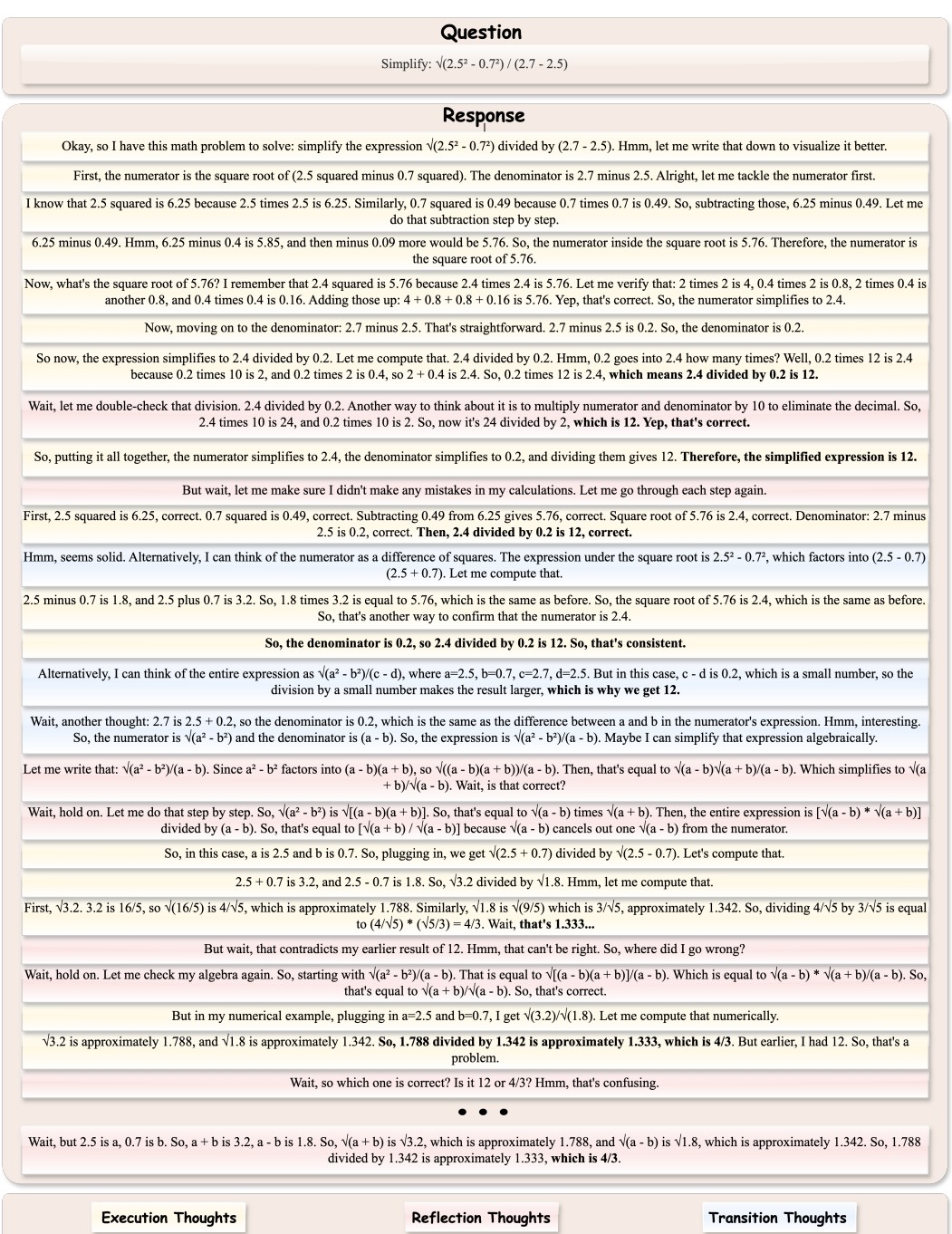

Figure 9: One example from DeepSeek-R1-Distill-Qwen-7B for a math question where the model fails to provide the correct answer after multiple reflections and transitions, even though it had arrived at the correct answer(**12**) multiple times before.

