# OpenReview forum: "SEAL: Steerable Reasoning Calibration of Large Language Models for Free"
_colmweb.org/COLM/2025/Conference — COLM 2025_

### Official Review · Reviewer_b7rX · 2025-05-10

**Rating:** 7
**Confidence:** 5
**Ethics Flag:** 1

**Summary:**

The paper proposes a way to calibrate the reasoning trace of o1 style reasoning models that usually produce long CoT chains to solve reasoning tasks. First they present evidence of the reasoning thoughts belonging to two categories: execution (ones involved in the step-by-step breakdown), and reflection and transition (ones where the model reflects on the steps and explores alternative approaches) using the hidden representations of the tokens separating thoughts (\n\n). They formulate a reasoning steering vector that encourages the models to execute more and reflect&transition less, which is computed by using a few 100 reasoning trace examples. During inference, the hidden representation of a thought is imputed with this steering vector, making the model obtain higher accuracy while utilizing fewer tokens. Overall, the paper is structured and reads well, presents a novel and computationally efficient approach to get these reasoning models to perform more efficiently.

**Questions To Authors:**

- Related to the mentioned weakness, what about for agentic tasks, how can this method be adopted?

**Reasons To Accept:**

- The paper is well-written and the topic is very relevant across domains
- The idea of grouping thoughts into the ones involved in the task breakdown and those for exploring alternate paths is not novel but the approach of doing it with just few examples is novel
- The offline computation of the steering vector involves a simple vector addition which is very neat
- This steering vector generalizes well across domains; at least the math and code that were tested

**Reasons To Reject:**

- Several works in the past have demonstrated how math and code domains benefit each other, so it would have been better to see this approach applied to a more distant domain rather than code using the steering vector from math. This would have been a better indicator of the generalizability of this approach

---

> ### Author Response · Authors · 2025-06-02
>
> Thanks for the constructive suggestion. We provide detailed results below.
>
> **More Benchmark Results**
>
> Thank you for the insightful question. As the DeepSeek–Distilled model used in our study does not currently support tool use, we chose to evaluate our method on a planning task [1], deliberately selecting a scenario with a significant domain gap between math and code. To assess cross-domain generalization, we continued using the steering vector extracted from the MATH dataset. As shown in Table R1, SEAL achieves a consistent improvement of 3.7% in accuracy, along with a 44% reduction in token usage. These results further validate the effectiveness and generalizability of our approach. We’d be happy to include additional benchmark evaluations if the reviewer has specific suggestions.
>
> Table R1. Results on the calendar planning task from the Natural Plan benchmark using Deepseek-Distill-7B.
>
> | Methods                | Accuracy (%) | Avg. Response Length |
> |----------------------|--------------|-----------------------|
> | Baseline                 | 14.9         | 6341                  |
> | SEAL                 | 18.6         | 3531                  |
>
> [1] NaturalPlan: Benchmarking LLMs on Natural Language Planning

---

> ### Author Response · Authors · 2025-06-08
>
> Dear Reviewer b7rX,
>
> Thanks for your valuable time. As the deadline for the discussion period is nearing, we would greatly appreciate it if you could kindly let us know whether our responses have adequately addresses your concern. And we are eager to engage in further discussions before the discussion period ends if you still have remaining concerns. Thank you very much for your time and efforts!
>
> Best wishes,
>
> Authors

---

> > ### Comment · Reviewer_b7rX · 2025-06-10
> >
> > Thanks authors for adding the planning task, it is great to see SEAL working well on this as well. I understand this will be included in the final paper if published, as expressed by my positive score.

---

### Official Review · Reviewer_ssbp · 2025-05-13

**Rating:** 6
**Confidence:** 4
**Ethics Flag:** 1

**Summary:**

This paper proposes SEAL (Steerable rEAsoning caLibration), a training-free method that improves the efficiency and accuracy of large language models (LLMs) in complex reasoning by identifying and calibrating flawed reasoning paths. SEAL extracts a reasoning steering vector in the latent space, which is then used to dynamically adjust hidden states for more efficient and accurate reasoning.

**Reasons To Accept:**

-	The statistical analysis of the three types of thoughts (execution, reflection, and transition) with task difficulty and the analysis of latent representations are insightful. Key observations (e.g., “frequent reflection and transition thoughts consume a significant token budget”, “execution thoughts are clearly separable from non-execution thoughts in the latent space”) could inspire future research on understanding LLM reasoning.
-	The proposed framework SEAL for controlling the ratio of reflection and transition thoughts is promising.  Experiments with three models (Deepseek-R1-distill-Qwen-1.5B, Deepseek-R1-distill-Qwen-7B, QwQ-32B-Preview) on three datasets (Math500, GSM8k, LiveCodeBench) demonstrate improvements in both accuracy and efficiency.

**Reasons To Reject:**

-	The categorization of the three types of thoughts is based on keyword identification, which is rather coarse. The table includes only a limited set of keywords, potentially leading to misclassification. It remains unclear how this affects the performance and reliability of SEAL in reasoning steering.
-	Some claims in the paper lack sufficient justification and would benefit from further clarification. For example, in lines 145-146, “This aligns with the expectation that shallow layers primarily capture low-level features, while deeper layers encode more abstract conceptual and semantic knowledge”. It’s not clear how this observation explains the separation of different thought types in deeper layers. Does the model rely on distinct features to generate each type of thought?
-	The idea of computing the reasoning steering vector as an arithmetic combination of different thought vectors lacks clear intuition. How does this approach selectively mitigate “unnecessary” reflection and transition thoughts while preserving “essential” execution thoughts? As presented, the method seems to generally penalize reflection and transition thoughts, without a mechanism to distinguish between useful and redundant instances.
-	There is a typo in line 198—“four datasets”. Only three datasets are used in the experiments.

---

> ### Author Response · Authors · 2025-06-02
>
> We’re glad that Reviewer ssbp acknowledged our work as “insightful” and “promising”, To address Reviewer ssbp’s concerns, we provide detailed responses in the following.
>
> **Q1: Robust Thinking Step Identification Method**
>
> In this work, we adopted a keyword-based identification method, as we found that reflection and transition steps in the thinking model exhibit relatively distinct patterns, making simple keyword matching both effective and efficient. That said, more robust methods, such as using a large language model (LLM) for step classification are certainly feasible.
>
> Additionally, we experimented with LLM-based identification and observed an 87.15% agreement with our keyword-based method. While the LLM occasionally misclassifies execution steps as reflection or transition, it still provides a strong general-purpose alternative. To evaluate its effectiveness, we extracted new steering vectors using LLM-based annotations and applied them in our method. As shown in Table R1, SEAL with LLM-based identification achieves performance improvements comparable to the one obtained with keyword-based matching.
>
> Table R1. Performance on MATH500 using Deepseek-distill-7B with different step identification methods.
>
> | Method                     | Accuracy (%) |
> |----------------------------|----------|
> | Baseline                   | 85.8     |
> | SEAL with Keyword | 89.4     |
> | SEAL with LLM        | 88.6     |
>
> Additionally, we’d like to emphasize that our mean-diff approach for vector extraction is inherently robust to light misclassification. As long as the positive samples (those containing the target step) are dominated by the correct category, and the negative samples contain few such instances, the resulting direction remains meaningful. Naturally, perfect classification would yield even higher-quality vectors, but our method is designed to tolerate some level of noise. In Appendix B.2, we conduct a keyword ablation study where we intentionally introduce misclassifications. While this slightly degrades vector quality, we are still able to extract effective steering directions. These findings further demonstrate the robustness of our approach to moderate classification noise.
>
> **Q2: Clarification of Lines 145–146**
>
> The statement, “This aligns with the expectation that shallow layers primarily capture low-level features, while deeper layers encode more abstract conceptual and semantic knowledge,” suggests that deeper layers are more likely to represent features associated with high-level semantics, as opposed to the token-level patterns captured by shallow layers. Since reflection and transition thoughts are inherently semantic in nature, it follows that the corresponding activation patterns are more salient in deeper layers. The visualization results in Figure 3 support this perspective and help explain why mid-to-late layers in our experiments tend to produce higher-quality directions for steering reflection and transition behaviors.
>
> **Q3: Selective Reasoning Calibration**
>
> Thanks for the question. We would like to clarify that our method preserves essential reflection and transition steps when the model exhibits a strong tendency toward them. By applying a moderate steering strength, the method primarily suppresses unnecessary or marginal cases when the hidden representations lie near the boundary between execution and reflection/transition thoughts. When the hidden representations are far from this boundary, the model continues to perform necessary reflection and transition behaviors as originally intended. Our goal is not to eliminate all reflection and transition, but rather to reduce redundant instances.
>
> For example, in Table R2, we compare the average number of reflection/transition steps before and after applying SEAL. For correctly answered samples where redundancy is low, SEAL leads to only a modest reduction (3.43 steps). In contrast, for incorrectly answered samples, which exhibit more redundant reflection/transition behavior, SEAL reduces the number of steps by a significant average of 55.43. This demonstrates that our method selectively suppresses redundant behaviors without impairing effective reasoning.
>
> Table R2. Average number of reflection/transition steps before and after applying SEAL.
>
> | Sample Type           | Before SEAL | After SEAL |
> |------------------------|-------------|------------|
> | Originally correct      | 8.74        | 5.31       |
> | Originally incorrect    | 69.73       | 14.30      |
>
> While we believe our **lightweight, training-free** method is practical and effective, we agree that a more adaptive steering mechanism would be a promising direction. Such adaptability could potentially be achieved through reinforcement learning, and we consider this a promising direction for future work.
>
> **Q4: Typo**
>
> Thank you for pointing it out. We will correct the typo in the revised version.

---

> > ### Comment · Reviewer_ssbp · 2025-06-09
> >
> > Thank you for your responses, which addressed most of my concerns. It would be great to see the LLM-based identification method incorporated into the paper. Overall, I recognize the value of this work and have increased my rating accordingly.

---

> ### Author Response · Authors · 2025-06-08
>
> Dear Reviewer ssbp,
>
> Thanks for your valuable time. As the deadline for the discussion period is nearing, we would greatly appreciate it if you could kindly let us know whether our responses have adequately addresses your concern. And we are eager to engage in further discussions before the discussion period ends if you still have remaining concerns. Thank you very much for your time and efforts!
>
> Best wishes,
>
> Authors

---

### Official Review · Reviewer_goZm · 2025-05-18

**Rating:** 6
**Confidence:** 3
**Ethics Flag:** 1

**Summary:**

This paper presents SEAL, a novel, training-free approach for calibrating reasoning in large language models (LLMs) by steering their internal representations to suppress redundant "reflection" and "transition" thoughts during chain-of-thought (CoT) reasoning. The method is motivated by empirical observations that excessive non-execution reasoning steps correlate with reduced model performance and increased inference cost.

**Questions To Authors:**

1. How sensitive is SEAL to the quality and domain of the dataset used to extract the steering vector?

2. Could more robust, possibly learned classifiers improve thought type identification beyond keyword matching?

**Reasons To Accept:**

1. The paper identifies a meaningful problem: redundancy in chain-of-thought (CoT) reasoning in LLMs, particularly due to excessive "reflection" and "transition" thoughts. This is well-motivated with empirical data.

2. Demonstrated across multiple datasets (Math500, GSM8K, LiveCodeBench) and models, SEAL consistently improves accuracy while reducing token usage significantly.

3. The steering vector extracted from one task generalizes to others, which suggests robustness and practical utility without task-specific retraining.

**Reasons To Reject:**

1. The method heavily relies on heuristic identification of thought categories and the assumption that their latent representations are separable. There is minimal theoretical grounding or formal analysis of why the steering vector should work as described.

2. While SEAL demonstrates performance improvements, it's unclear whether the gains stem from suppressing redundant thoughts or from simply shortening sequences (which may reduce token-level errors). There is no disentangled analysis isolating these effects.

3. The method depends on specific hyperparameters (e.g., the steering layer and coefficient α), but the ablation study is shallow and lacks statistical robustness (e.g., no confidence intervals or multiple seeds). It is unclear how stable the results are across different configurations.

---

> ### Author Response · Authors · 2025-06-02
>
> Thank you for the positive evaluation and for acknowledging that our work “identifies a meaningful problem” and that our method shows “robustness and practical utility.” We provide point-by-point responses below.
>
> **Q1: Theoretical Justification of Steering Methods**
>
> Thanks for pointing it out. The theoretical analysis of activation-based intervention methods remains an open research challenge. Existing theoretical work on neural networks is often limited to small-scale models and simplified settings (e.g., omitting positional encodings), which makes it difficult to extend such analyses to large language models (LLMs). Moreover, reasoning behavior in LLMs involves complex interactions across multiple tokens, further complicating theoretical justification. For these reasons, we consider a full theoretical treatment beyond the scope of this work.
>
> That said, there is strong intuitive and empirical support for our approach. A growing body of interpretability research has shown that LLM activations encode meaningful semantic directions. For instance, prior work [1–3] demonstrates that specific features, such as concepts, refusals, or reasoning patterns, can be extracted from activation vectors, and that manipulating these directions can systematically influence model behavior. In our study, we observe that similar activation directions correspond to reflection and transition behaviors, and we show that modifying these directions leads to measurable changes in reasoning performance.
>
> **References**
>
> [1] Scaling Monosemanticity: Extracting Interpretable Features from Claude 3 Sonnet
>
> [2] Refusal in Language Models Is Mediated by a Single Direction
>
> [3] AxBench: Steering LLMs? Even Simple Baselines Outperform Sparse Autoencoders
>
> **Q2: Comparison with Simply Shortening Sequences**
>
> Great question! To isolate the effect of SEAL from simple response shortening, we conducted a comparison with a baseline method that truncates the thinking process. Specifically, we introduced a fixed “think budget,” forcing the model to insert an end-of-think token (</think>) once the generated thinking sequence exceeds the budget. This transition moves the model into the summarization phase, resulting in a final answer. By setting the reasoning budget to 3500 tokens, we aligned the average response length with that of SEAL for a fair comparison.
>
> As shown in Table R1, this truncation-based approach underperforms SEAL, despite producing similarly shorter outputs. This indicates that SEAL’s advantage stems not merely from shortening the response, but from guiding the model to reduce unnecessary reflection and transition steps, effectively altering its reasoning behavior.
>
> Table R1. Comparison between SEAL and simple truncation on Deepseek-distill-7B (MATH500).
>
> | Method                      | Accuracy (%) | Avg. Response Length |
> |----------------------------|--------------|-----------------------|
> | Baseline                       | 85.8         | 3389                  |
> | Baseline + Think Budget = 3500 | 85.0         | 2540                  |
> | SEAL                       | 89.4         | 2661                  |
>
> **Q3: Robustness of the Ablation Study**
>
> In our experiments, we used a fixed steering coefficient of 1.0 across all tasks and models, as this value consistently delivered strong performance across diverse settings. To assess robustness, we repeated the ablation study over three runs and observed stable results, with low variance in accuracy. The average performance and standard deviation for different coefficients and layers are reported in Table R2 and  Table R3.
>
> Table R2. Ablation results on steering coefficient using Deepseek-Distill-7B on the MATH500 dataset.
> | Coefficient | Accuracy (%) ± Std |
> |-------------|---------------------|
> | 0.0         | 86.07 ± 0.31        |
> | 0.5         | 88.27 ± 0.64        |
> | 1.0         | 89.40 ± 0.20        |
> | 1.25        | 88.87 ± 0.12        |
> | 1.5         | 88.07 ± 1.80        |
> | 2.0         | 88.00 ± 0.40        |
>
> Table R3. Ablation results on the target steering layer using Deepseek-Distill-7B on the MATH500 dataset.
> | Layer Index | Accuracy (%) ± Std |
> |-------------|---------------------|
> | 5           | 86.40 ± 0.35        |
> | 10          | 85.73 ± 0.42        |
> | 15          | 86.33 ± 0.90        |
> | 20          | 89.40 ± 0.20        |
> | 25          | 89.27 ± 0.50        |
> | 28          |86.63    ± 0.23               |

---

> > ### Author Response · Authors · 2025-06-02
> >
> > **Q4: Steering Vector Across Domains**
> >
> > Thanks for the question. First, our steering vector generalizes across domains because it captures reflection and transition-related behaviors rather than task-specific features. As shown in Table 2, applying a vector extracted from a math task to a code task still yields strong performance, indicating the domain-agnostic nature of the signal.
> > Second, we also conducted experiments using steering vectors extracted from various tasks, such as GSM8K and Code. These results, presented in Appendix C, demonstrate consistently strong performance across domains, further validating the robustness and transferability of our approach to vector extraction.
> >
> >
> > **Q5: Other Step Identification Method**
> >
> > In this work, we adopt a keyword-based identification method, as we observed that reflection and transition steps in the thinking model tend to follow easily recognizable patterns. As a result, simple keyword matching proves to be both effective and efficient.
> >
> > That said, more robust approaches, such as using a large language model (LLM) for step identification, are certainly viable. We experimented with an LLM-based method and found it achieves a high 87.15% agreement with our keyword-based approach.
> >
> > We further tested the effectiveness of LLM-based identification by extracting new steering vectors using this method. As shown in Table R4, applying steering based on LLM-identified steps still leads to a notable performance improvement.
> >
> > Table R4. Performance on MATH500 using different step identification strategies with Deepseek-distill-7B.
> >
> > | Method                     | Accuracy (%) |
> > |----------------------------|----------|
> > | Baseline                   | 85.8     |
> > | SEAL with Keyword | 89.4     |
> > | SEAL with LLM        | 88.6     |
> >
> > An important point to emphasize is that our mean-diff vector extraction method is inherently robust to minor misclassifications. As long as the positive samples (i.e., samples containing the target step) predominantly reflect the intended behavior, and the negative samples do not, the extracted direction remains meaningful. Naturally, more accurate classification yields higher-quality vectors, but perfect labeling is not a strict requirement.
> >
> > In **Appendix B.2**, we perform an ablation on the keyword set and deliberately introduce misclassifications. While this results in some degradation in vector quality, we still extract effective steering directions, further demonstrating that our method tolerates a reasonable level of noise in the step identification process.

---

> > > ### Comment · Reviewer_goZm · 2025-06-05
> > > **Response to the Rebuttal**
> > >
> > > Thank you for your detailed response. Your clarifications have addressed the majority of my concerns, and I appreciate the effort in explaining the results. I hope these findings can be incorporated into a future revision of the paper. That said, I believe my initial evaluation remains appropriate, and I would prefer to retain my original score. Overall, I consider this paper to be **above the acceptance threshold**.

---

### Official Review · Reviewer_hU5P · 2025-05-26

**Rating:** 7
**Confidence:** 4
**Ethics Flag:** 1

**Summary:**

This paper proposes SEAL, a new method to reduce redundant reasoning in LLMs without additional training. The authors find that CoT often include unnecessary reflection and transition steps that waste tokens and can mess up reasoning. SEAL computes a steering vector that shifts the model’s hidden states toward the execution direction. During generation, this vector is added to the hidden state after each thought, steering the model to focus on direct problem-solving steps. The approach is training-free and model-agnostic. SEAL can improve accuracy by up to 11\% and cut reasoning tokens by 12–50\% on math and coding benchmarks.

**Questions To Authors:**

1. Can SEAL mis-steer the reasoning process? For example, could the model skip necessary verification steps when reflection is suppressed? How does the method detects and handles the situations where injecting the steering vector harms performance?
2. The paper fixed a steering coefficient. How was this value chosen? How sensitive are the results to this parameter?

**Reasons To Accept:**

1. Insightful analysis of reasoning patterns: The paper identifies three intuitive reasoning thought categories (execution, reflection, transition) and empirically shows that failure cases have substantially more reflection/transition steps than successes.
2. Latent space insights: The work demonstrates that the three thought types are separable in latent space (especially in deeper transformer layers).
3. Training-free intervention: SEAL method effectively steers hidden states using a learned vector offset, rather than fine-tuning model's weights. This representation-level intervention is new and avoids additional training cost or architectural changes.
4. Performance improvements: SEAL leads to notable improvements on complex reasoning tasks.

**Reasons To Reject:**

1. Heuristic categorization: The method identifies reflection vs. transition thoughts by detecting keywords. This may limit generality for different domains or writing styles. A more automated or learned way to label thought types could strengthen the approach.
2. Diminishing returns on advanced models: The gains of SEAL decrease with very capable models such as the case of the 32B model. This suggests that powerful models already manage their reasoning well, and steering can overshoot. The authors may want to discuss when SEAL might hurt performance or how to detect such cases.
3. Balanced steering: While SEAL aims to reduce reflection, the results indicate not all reflection is bad. The authors also need to tune a steering strength to avoid eliminating too much. This fine balance might be tricky to find for each model/task.
4. Scope of benchmarks: All experiments are on math or coding tasks with structured problem-solving. It remains to be seen how SEAL performs on more open-ended reasoning such as commonsense QA or planning.

---

> ### Author Response · Authors · 2025-06-02
>
> We appreciate Reviewer hU5P’s positive initial evaluation and are grateful for the recognition of our work as providing “insightful analysis,” introducing “new” methods, and achieving “notable improvements.” To address the reviewer’s concerns, we provide point-by-point responses below.
>
>
> **Q1: Generalization of Thought Labeling Methods**
>
> Thank you for the question. We adopt a keyword-based identification approach, as we observed that reflection and transition steps in the reasoning process tend to follow relatively distinct and recognizable patterns. These can be effectively captured through simple keyword matching.
> To support generalization across different domains or writing styles, large language models (LLMs) can also be used to identify thought types. In our analysis, the keyword-based method showed a high degree of consistency with LLM-based identification, achieving an 87.1% agreement. Furthermore, the end-to-end performance improvements achieved using keyword matching are similar to those obtained with LLM-based identification. This demonstrates the robustness and generalizability of our approach across different classification strategies.
>
> **Results**: As shown in **Table R1**, both identification methods lead to significant improvements over the baseline full model, with accuracy gains ranging from 2.80% to 3.60%.
>
> Table R1. End-to-end performance with different thought identification methods. Evaluated on Deepseek-Distill-7B and the MATH500 test set.
> | Method                     | Accuracy |
> |----------------------------|----------|
> | Baseline                   | 85.8     |
> | SEAL with Keyword | 89.4     |
> | SEAL with LLM        | 88.6     |
>
>
> **Q2: Improvements on Advanced Models**
>
> Thanks for pointing it out. First, we would like to clarify that our method is not limited to small models. The QWQ-32B-preview model (used in our manuscript) is an early-stage reasoning model that has not undergone large-scale reinforcement learning (RL) training, which explains why the over-reflection and over-transition phenomena are less prominent. Furthermore, we evaluated our method on the newly released QWQ-32B model, which has been extensively trained with RL. Our method remains highly effective on this more advanced model, as shown in **Table R2**
>
> Table R2. Performance on QWQ-32B across Math500 and LiveCodeBench datasets.
>
> | Dataset     | Method | Accuracy | Avg. Token Count |
> |----------------|--------|----------|------------------|
> | Math500    | Baseline   | 91.1     | 3650             |
> | Math500    | SEAL        | 92.5     | 3160             |
> | LiveCodeBench | Baseline | 76.2     | 5610             |
> | LiveCodeBench | SEAL   | 82.8     | 4959             |
>
> Second, it is indeed the case that over-reflection and over-transition behaviors are more pronounced in smaller models. However, smaller reasoning models remain widely used in practice due to resource constraints. The fact that our method effectively mitigates these behaviors in small models makes it especially valuable and practical for a broader range of use cases. Therefore, we believe our method holds strong practical value for improving the performance and efficiency of current reasoning models.

---

> > ### Author Response · Authors · 2025-06-02
> >
> > **Q3: Balanced Steering and Preventing Mis-Steering**
> >
> > We would like to clarify that our method does not aim to eliminate all reflection and transition steps. Instead, we apply a moderate steering coefficient (1.0 in our experiments) to reduce unnecessary or marginal reflection/transition behavior. When the model exhibits a strong tendency to reflect or transition, it is still permitted to do so in order to avoid mis-steering.
> >
> > For example, **Table R3** compares the average number of reflection/transition steps before and after applying SEAL. For samples that the model answers correctly, the number of reflection/transition steps is relatively low, indicating minimal redundancy. In these cases, SEAL makes only slight reductions (3.43 steps). In contrast, for incorrectly answered samples, where reflection/transition behaviors are more excessive, SEAL significantly reduces the number of such steps (55.43 steps). This demonstrates that the method effectively suppresses redundant behaviors without interfering with genuinely helpful reasoning processes.
> >
> > Table R3. Average number of reflection/transition steps before and after applying SEAL.
> > |  Sample Type        | Before SEAL | After SEAL |
> > |--------------------|-------------|------------|
> > | Originally correct  | 8.74        | 5.31       |
> > | Originally incorrect | 69.73       | 14.30      |
> >
> > Additionally, we agree that a more adaptive steering strategy would be ideal. Such adaptation could potentially be achieved through reinforcement learning. However, our current goal is to propose a **lightweight, training-free method**. Developing adaptive, RL-based approaches is a promising direction we plan to explore in future work.
> >
> >
> > **Q4: Additional Benchmark on Planning**
> >
> > Good point. We conducted additional experiments on a planning task using the Natural Plan dataset [1], while still leveraging the reasoning vectors extracted from MATH to demonstrate their generalizability. As shown in Table R4, our method yields notable improvements in both accuracy and response length, demonstrating its effectiveness and generalizability. We will include these results in the updated draft.
> >
> > Table R4. Results on the calendar planning task from the Natural Plan benchmark using Deepseek-Distill-7B.
> > | Methods                | Accuracy (%) | Avg. Response Length |
> > |----------------------|--------------|-----------------------|
> > | Baseline                 | 14.9         | 6341                  |
> > | SEAL                 | 18.6         | 3531                  |
> >
> > [1] NaturalPlan: Benchmarking LLMs on Natural Language Planning
> >
> >
> > **Q5: Steering Coefficient Selection**
> >
> > Thank you for the question. We provide a detailed ablation study on the steering coefficient in Section 5.3.3, where the model demonstrates consistently strong performance with $\alpha = 1.0$  across different models and tasks. Therefore, we use a fixed coefficient throughout all experiments to avoid task-specific tuning.

---

> ### Author Response · Authors · 2025-06-08
>
> Dear Reviewer hU5P,
>
> Thanks for your valuable time. As the deadline for the discussion period is nearing, we would greatly appreciate it if you could kindly let us know whether our responses have adequately addresses your concern. And we are eager to engage in further discussions before the discussion period ends if you still have remaining concerns. Thank you very much for your time and efforts!
>
> Best wishes,
>
> Authors

---

### Decision · Program_Chairs · 2025-07-08

**Decision:**

Accept

**Comment:**

This paper first analyzes CoT reasoning traces and found that the thoughts (identified using \n\n boundaries) can be divided into three types: execution, reflection, and transition. They also showed that in hidden states these different types are also distinguishable using tSNE. Their analyses also found that redundant reasoning primarily results from unnecessary reflection and transition steps. To improve reasoning efficiency, the authors introduce a training-free approach that uses a steering vector added after each thought block during inference, encouraging LLMs toward execution-oriented reasoning. Experiments demonstrate the steering vector improves both accuracy and efficiency, and even generalizes across tasks.

Pros:
1. Reviewers find the categorization of CoT reasoning into execution, reflection, and transition, and that they are distinguishable in hidden states, to be insightful.
2. The proposed approach, being training-free, improves performance in terms of both accuracy and efficiency without requiring fine-tuning.

Cons:
1. As pointed out by a reviewer, the gain seems to diminish on advanced models, which might already be doing reasoning well and steering is less helpful.
2. The approach requires specifying hyperparameters, notably the layer and steering coefficient. Results from the authors during rebuttal (Table R3) indicate sensitivity to the chosen layer, potentially limiting practicality.

Overall, reviewers positively rate this paper (with scores of 7, 6, 6, 7), especially regarding the insightful categorization of CoT reasoning and the effective training-free steering approach to promote execution. Therefore, I recommend accepting this paper.